# m^6^A Methylases Regulate Myoblast Proliferation, Apoptosis and Differentiation

**DOI:** 10.3390/ani12060773

**Published:** 2022-03-18

**Authors:** Xinran Yang, Chugang Mei, Xinhao Ma, Jiawei Du, Jianfang Wang, Linsen Zan

**Affiliations:** 1College of Animal Science and Technology, Northwest A & F University, Xianyang 712100, China; yangxinran93@nwafu.edu.cn (X.Y.); meichugang@163.com (C.M.); maxinhao@nwafu.edu.cn (X.M.); dujiawei@nwafu.edu.cn (J.D.); jfwang@nwafu.edu.cn (J.W.); 2National Beef Cattle Improvement Center, Northwest A & F University, Xianyang 712100, China

**Keywords:** *N*^6^-methyladenosine, m^6^A methylases, proliferation, apoptosis, myoblast differentiation, cattle, myogenesis

## Abstract

**Simple Summary:**

*N*^6^-methyladenosine (m^6^A) is the most prevalent methylation modification in eukaryotic mRNA, and it plays an important role in regulating gene expression. Previous studies found that m^6^A methylation plays a role in mammalian skeletal muscle development. Skeletal muscle is an important factor that regulates livestock muscle quality and maintains metabolic homeostasis, and skeletal myogenesis is regulated by a series of transcription factors. However, the role of m^6^A in bovine skeletal myogenesis is unclear. In this study, we examined the expression patterns of the m^6^A methylase genes *METTL3*, *METTL14*, *WTAP*, *FTO* and *ALKBH5* in bovine skeletal muscle tissue and during myogenesis in myoblasts. Furthermore, we used bovine skeletal muscle myoblasts as the object of study to discover the regulatory role of these genes in the process of skeletal myogenesis in vitro. Our findings indicate that these five m^6^A methylases have pronounced and diverse functions in regulating bovine skeletal myoblast proliferation, apoptosis and myogenic differentiation, which can contribute to further understanding the roles of m^6^A in skeletal muscle development.

**Abstract:**

*N*^6^-methyladenosine (m^6^A) plays an important role in regulating gene expression. Previous studies found that m^6^A methylation affects skeletal muscle development. However, the effect of m^6^A methylases on bovine skeletal myogenesis is still unclear. Here, we found that the expression of m^6^A demethylases (*FTO* and *ALKBH5*) was significantly higher in the longissimus dorsi muscle of adult cattle than in newborn cattle. In contrast, the expression of m^6^A methyltransferases (*METTL3*, *METTL14* and *WTAP*) was reduced. The mRNA expression of all five genes was found to be increased during the myogenesis of myoblasts in vitro. Knockdown of FTO or METTL3 promoted myoblast proliferation, inhibited myoblast apoptosis and suppressed myogenic differentiation, whereas ALKBH5 knockdown had the opposite effect. METTL14 knockdown enhanced myoblast proliferation and impaired myogenic differentiation. WTAP knockdown attenuated proliferation and contributed to apoptosis but did not affect differentiation. Furthermore, the functional domains of these five m^6^A methylases are conserved across species. Our results suggest that m^6^A methylases are involved in regulating skeletal muscle development and that there may be a complex network of m^6^A methylation regulating skeletal myogenesis.

## 1. Introduction

The growth and development of skeletal muscle is crucial for meat production and the quality of livestock. On the one hand, the weight of skeletal muscle directly determines the meat yield of animals; on the other hand, the content of intramuscular fat and nutrients in skeletal muscle are closely related to meat quality [1,2]. In addition, skeletal muscle is an important motor and energy metabolizing tissue of the body that plays an important role in maintaining metabolic balance and homeostasis in the body. Therefore, it is essential to investigate the growth and development patterns of skeletal muscle to improve the production performance of meat animals and to explore their muscle physiopathology. Skeletal myogenesis is a very complex biological process that includes the stages from muscle satellite-cell activation to myoblast proliferation and, finally, to terminal differentiation. In addition to the regulation of this process by a series of key transcription factors, some epigenetic factors such as DNA methylation and histone modification, also play vital roles [3]. The role and molecular mechanism of m^6^A (*N*^6^-methyladenosine) RNA methylation modification in the growth and development of bovine skeletal muscle are still unclear.

In eukaryotic mRNAs, m^6^A is the most common methylation modification [4,5], and methylation modification has been shown to be dynamically reversible and is catalyzed by METTL3 (methyltransferase 3), METTL14 (methyltransferase 14) and WTAP (Wilms’ tumor 1-associated protein), which comprise the methyltransferase complex (MTC) that facilitates m^6^A methylation of adenosine in RNA [6,7]. Meanwhile, m^6^A demethylases, including FTO (fat mass and obesity-associated protein) and ALKBH5 (AlkB homolog 5), demethylate adenine that has undergone m^6^A modification [8,9], which tends to be functionally bound and characterized by m^6^A reader proteins such as the YTH domain family proteins and the hnRNP family proteins [10]. There is growing evidence that m^6^A modifications play important biological functions in regulating cell fate, the cell cycle, embryonic stem cell reprogramming, and individual development by modulating RNA stability, localization, transport, shearing and translation at the post-transcriptional level [10,11,12,13].

In recent years, with the development of m^6^A-seq technology, studies have gradually revealed the m^6^A methylation modifications occurring in farm animals. A growing number of studies, including our own, revealed the potentially important role of m^6^A modification in muscle and fat growth and development in pigs, chickens, cattle and goat through m^6^A transcriptome profiling [14,15,16,17,18,19]. However, studies on m^6^A methylation modification in bovine skeletal muscle development are rarely reported, and its function and mechanism in regulating myoblast proliferation and differentiation in vitro remain unclear. In this study, we examined the expression patterns of the m^6^A methylase genes *METTL3*, *METTL14*, *WTAP*, *FTO* and *ALKBH5* in bovine skeletal muscle tissue and during myogenesis in myoblasts. The roles of these five well-known m^6^A methylases in regulating myoblast proliferation, apoptosis and differentiation in vitro were further analyzed by loss-of-function and gain-of-function assays. Our findings can provide a new perspective for genetic improvement and molecular breeding in beef cattle and serve as a reference for studying the mechanisms of RNA methylation modification in skeletal myogenesis.

## 2. Materials and Methods

### 2.1. Culture and Differentiation of Bovine Myoblasts

The longissimus dorsi muscle of three newborn (1-day-old) and three adult cattle (24-month-old) as well as the myoblasts used in this study were previously preserved in our laboratory [19,20]. Myoblasts were cultivated in growth medium until they reached 80–90 percent confluence, and then differentiation medium was used to induce myogenic differentiation. The culture conditions were a humidified incubator (Thermo Fisher Scientific, Waltham, MA, USA) with 5% CO_2_ at 37 °C. DMEM/F12 (BI, Watertown, MA, USA) was used as the myoblast growth medium, which contained 20% FBS (GIBCO, Grand Island, NY, USA) and 1% penicillin/streptomycin. The myoblast differentiation medium consisted of 97% DMEM/F12, 2% horse serum (BI) and 1% penicillin/streptomycin. The medium was replaced every two days.

### 2.2. RNA Isolation, cDNA Synthesis and Real-Time Quantitative PCR

Muscle tissue and myoblasts were harvested, and total RNA was isolated using RNAiso reagent (TaKaRa, Dalian, China). The PrimeScript RT reagent kit (TaKaRa) was used to synthesize cDNA. Real-time quantitative PCR (RT-qPCR) was performed using the TB Green Premix Ex Taq II Kit (TaKaRa) and a CFX Connect qPCR Detection System (BIO-RAD, CA, USA). To standardize the results, *GAPDH* was employed as an internal reference. Relative mRNA expression was calculated using the 2^−ΔΔCt^ method [21].

The differentiation status of myoblasts was evaluated by detecting the mRNA levels of *MYOD1* (myogenic differentiation 1), *MYOG* (myogenin), *MYF6* (myogenic factor 6, also known as MRF4), *MYH3* (myosin heavy chain 3), *MYMK* (myomaker, myoblast fusion factor) and *CKM* (creatine kinase), which are widely recognized marker genes of differentiated myoblasts and fused myotubes [22,23]. To investigate the potential role of m^6^A methylases in bovine skeletal myogenesis, we selected the five most well-known m^6^A methylases, including m^6^A methyltransferases METTL3, METTL14, WTAP and m^6^A demethylases FTO and ALKBH5 and examined their mRNA expression patterns in bovine skeletal muscle and myoblasts. All primers used in RT-qPCR are listed in Appendix A.

### 2.3. RNA Interference, Plasmid Construction and Transfection

Genepharma (Shanghai, China) synthesized all siRNAs for this study, and the sequences are shown in Appendix A. The coding sequences (CDS) of bovine *FTO* (NM_001098142), *ALKBH5* (NM_001205517.3), *METTL3* (NM_001102238), *METTL14* (NM_001083714.1) and *WTAP* (NM_001113254.2) were synthesized via PCR and cloned into the pcDNA3.1(+) expression plasmid. All primers are listed in Appendix A. When the myoblasts reached 80–90% confluence, the cells were seeded in 6-well plates. The instantaneous transfection procedure was performed according to the protocol of the Lipofectamine 3000 transfection reagent (Invitrogen, Carlsbad, CA, USA), and three replicate wells were transfected each time.

### 2.4. EdU Staining

EdU staining experiments were performed according to the protocol of the Cell-Light EdU Apollo567 In Vitro Kit (RiboBio, Guangzhou, China). Briefly, the cells were inoculated in 48-well plates. Cell transfection was performed when cell density reached approximately 50%. After continuing incubation for 48 h, EdU solution was added to each well and incubated for 2 h. After fixation and permeabilization, Apollo staining solution was added, followed by incubation for 30 min. Cells were then washed with PBS 3 times and then incubated in 1X Hoechst33342 for 30 min to stain cell nuclei. Finally, 100 μL PBS was added to each well for immediate observation and photography under an Evos-fl-auto2 microscopy imaging system (Thermo Scientific, Waltham, MA, USA).

### 2.5. CCK-8 Assay

Cells were inoculated into 100 μL of culture medium in a 96-well plate to perform cell transfection when the cell density reached 50%. After 48 h of incubation, 10 μL of Cell Counting Kit-8 (CCK-8, TargetMOI, Boston, MA, USA) was added directly to the cell culture medium and mixed thoroughly. Incubation was continued for another 2 h and the absorbance at 450 nm was then read using a multimode plate reader (Infinite M200 Pro, TECAN, Männedorf, Switzerland).

### 2.6. Flow Cytometry (FCM) Assay

Cell-cycle assays were performed according to the protocol of the Cell Cycle Staining Kit (MultiSciences, Hangzhou, China). Cells were inoculated in 6-well plates and grown to 80–90% confluency, trypsin digested, and the supernatant removed by centrifugation. Propidium iodide (PI) (500 μL) and 5 μL permeabilization solution were added to each well, followed by incubation for 30 min at room temperature while protected from light. The cell counts were detected and analyzed on a flow cytometer (BD FACS AriIII, BD, Franklin Lakes, NJ, USA) at the Large Scale Instrument and Equipment Sharing Platform of Northwestern A & F University.

Apoptosis assays were performed according to the protocol of the Annexin V-FITC/PI Apoptosis Kit (MultiSciences). Apotosis Positive Control Solution (500 μL) was added to collected cells and incubated on ice for 30 min. They were then washed with PBS and resuspended with 500 μL pre-chilled 1X Binding Buffer with addition of 5 μL Annexin V-FITC and 10 μL PI and then incubated at room temperature in the dark for 5 min. After centrifugation and PBS resuspension, the cells were detected in the flow cytometer (BD FACS AriIII, BD).

### 2.7. Western Blotting

Cells were collected and lysed on ice in Western and IP cell-lysis buffer (Beyotime Biotechnology, Shanghai, China) containing 1% PMSF (Solabio, Beijing, China) for 30 min. The lysates were collected with a cell scraper and centrifuged at 14,000× *g* for 15 min at 4 °C to collect the protein from the supernatants. Then, the protein concentrations were determined using a BCA Protein Analysis Kit (Beyotime Biotechnology). All cell proteins were incubated at 100 °C for 10 min in SDS-PAGE sample buffer. The proteins were separated by SDS-PAGE and transferred to PVDF membranes for immunoblotting. The membranes were incubated with specific primary antibodies overnight at 4 °C and then incubated with secondary antibodies at room temperature for 2 h. Western blotting was performed using the chemiluminescence method (ECL Plus detection system), and the band intensities were quantified using the ImageJ software (NIH, Bethesda, MD, USA). The antibodies used in this study are shown in Appendix A.

### 2.8. Immunofluorescence

To verify that the cells used were capable of myogenic differentiation, we inoculated the cells in culture dishes. They were grown to 80–90% confluency before subculturing into 6-well plates for further culture. After 48 h of growth, the expression of MYOD1 (myogenic differentiation 1) and PAX7 (paired box 7) proteins was detected using immunofluorescence assays. When myoblasts grew again to 80% density, myogenic differentiation was induced, and on day 3 of differentiation, MYHC (myosin heavy chain) protein was visualized to reflect myotube formation. Briefly, the cells were washed with PBS, fixed with 4% paraformaldehyde at room temperature for 20 min, and then permeabilized with 0.2% Triton X-100 (Solarbio, Beijing, China) for 10 min. The cells were subsequently blocked with 0.3 M glycine, 10% donkey serum and 1% BSA in PBS at room temperature for 1 h. Then, the cells were incubated with primary antibodies overnight at 4 °C. After washing 3 times with PBS, the cells were incubated with fluorescent dye-conjugated secondary antibodies for 1.5 h at 37 °C and protected from light. The cells were washed 3 times with PBS, stained with 0.1% DAPI (Sigma-Aldrich, St. Louis, MO, USA) for 15 min and then visualized under Evos-fl-auto2 microscopy imaging system (Thermo Scientific) or a fluorescence microscope (Olympus IX71 (Olympus Corporation, Tokyo, Japan). The antibodies used in this study are shown in Appendix A.

### 2.9. Bioinformatics Analysis

The protein sequences for FTO, ALKBH5, METTL3, METTL14 and WTAP from humans, mice, pigs, chickens, goats, sheep, cattle and zebu cattle were each downloaded from the NCBI website. A phylogenetic tree was constructed using MEGA7 software, and the conservation of functional domains in these five proteins was subsequently analyzed using the EvolView online tool [24].

### 2.10. Statistical Analysis

All data are displayed as the means ± standard deviations (SD) of at least three biological replicates. Student’s *t*-test (between two groups) or ANOVA (among multiple groups) were used to compare the significance of the data. The significance levels *p* < 0.01 or *p* < 0.05 defined the differences as either very significant or significant, respectively. GraphPad Prism 7.00 (GraphPad Software, San Diego, CA, USA) software was used to analyze the results and produce images.

## 3. Results

### 3.1. Identification of Bovine Skeletal Myoblasts

The results showed that both MYOD1 and PAX7 were expressed, though MYOD1 expression was lower as indicated by weaker fluorescence (Figure 1A). After induction of myogenic differentiation, myoblasts gradually fused to form myotubes and continued to lengthen and enlarge (Figure 1B). Notably, the levels of *MYOG*, *MYF6, MYH3*, *MYMK* and *CKM* gradually increased during myogenic differentiation, while the levels of MYOD1 peaked on day 2 (Figure 1C). These findings were consistent with previous studies showing that MyoD1 plays a vital role in the proliferation and early differentiation of myoblasts [25,26]. These results suggest that the isolated bovine skeletal myoblasts could undergo myogenic differentiation and can be used as a model for our subsequent studies.

### 3.2. mRNA Expression of m^6^A Methylase Genes in Bovine Skeletal Muscle and Myoblasts

Briefly, m^6^A methylases were found to be important regulators of muscle growth and development and do so by mediating m^6^A methylation. We detected the expression of *METTL3*, *METTL14*, *WTAP*, *FTO* and *ALKBH5* in the longissimus dorsi muscle of both newborn and adult cattle. The results showed that the expression of *FTO* and *ALKBH5* was significantly higher in the longissimus dorsi muscle of adult cattle than in newborn cattle (Figure 2A). In contrast, *METTL3*, *METTL14* and *WTAP* were expressed at lower levels in the longissimus dorsi muscle of adult cattle than in newborn cattle (Figure 2A). We further examined the temporal expression profiles of these five genes during myoblast proliferation and differentiation in vitro. The results showed that their expression changed moderately during the proliferative phase of myoblasts, while all were elevated in the early stage of myogenic differentiation (Figure 2B–F). Among them, the expression of *FTO* and *METTL14* increased during myoblast proliferation, while the expression of *ALKBH5* and *METTL3* decreased, and the expression of *WTAP* increased in the prophase of proliferation and then decreased (Figure 2B–F). Moreover, the expression of *FTO*, *METTL3* and *METTL14* increased in the early stage of myogenic differentiation and decreased in the middle and late stages, while the expression of *ALKBH5* and *WTAP* increased gradually with the progression of differentiation (Figure 2B–F). Our results suggest a potential role for FTO, ALKBH5, METTL3, METTL14 and WTAP in skeletal myogenesis, implying that m^6^A methylation may be involved in the regulation of bovine skeletal muscle growth and development.

### 3.3. m^6^A Methylases Regulate Myoblast Proliferation

To investigate whether these five m^6^A methylases affected myogenesis, we synthesized siRNAs against them and selected those with the highest interference efficiency for subsequent experiments (Appendix A). EdU staining analysis showed that knockdown of FTO, METTL3 and METTL14 each increased the percentage of EdU-positive cells, while ALKBH5 and WTAP knockdown decreased the percentage of EdU-positive cells (Figure 3A,B). However, the CCK-8 assay revealed that only FTO knockdown promoted myoblast proliferation, whereas the other gene knockdowns did not affect the change in the number of proliferating cells (Figure 3C). Using flow cytometry, we found that FTO knockdown did not change the number of cells in different phases of the cell cycle; ALKBH5 knockdown suppressed the numbers of cells in the S phase, while METTL3 knockdown increased the number of cells in the S phase (Figure 3D,E). Meanwhile, with the knockdown of METTL14 and WTAP, the number of cells in the G1 phase decreased, while the number of cells in the G2 phase increased (Figure 3D,E). Moreover, we examined the expression of crucial cell-cycle factors (Figure 3F). Knockdown of FTO, METTL3 or METTL14 all promoted the mRNA expression of *CCNA2*, *CCNE1* and *PCNA*, but METTL3 knockdown did not affect *CDK1* expression, and METTL14 knockdown did not alter the expression of *MCM6* (Figure 3F). Alternatively, ALKBH5 or WTAP knockdown generally suppressed the expression of these cell-cycle genes. Nevertheless, among them, the expression of *CCNE1* was not altered by ALKBH5 knockdown, and WTAP knockdown did not change the expression of *CCNA2* and *PCNA* (Figure 3F). In addition, Western blot assays illustrated that the knockdown of FTO obviously promoted the expression of CCNB1, CCNE1 and CDK1 proteins, and METTL14 knockdown promoted the expression of CCNB1 and CDK1 proteins (Figure 3G). ALKBH5 knockdown clearly inhibited CDK1 and PCNA protein expression. METTL3 knockdown promoted the expression of CCNB1 but was not expected to promote the expression of p21 protein, a cell-cycle inhibitory protein. WTAP knockdown inhibited CCNB1 and CDK1 protein levels while promoting the expression of p21 (Figure 3G). All these results indicate that FTO, METTL3 and METTL14 knockdown all promoted myoblast proliferation, which was inhibited by ALKBH5 and WTAP knockdown. It also revealed that they have different effects on cell-cycle factors, the mechanism of which needs to be further explored.

### 3.4. m^6^A Methylases Regulate Myoblast Apoptosis

While exploring the effect of m^6^A methylases on myoblast proliferation, we investigated the effect of these five m^6^A methylases on myoblast apoptosis. The results of apoptosis detection by flow cytometry demonstrated that FTO, ALKBH5 and WTAP knockdown increased the number of apoptotic cells, while METTL3 knockdown caused a decrease and METTL14 knockdown had no significant effect (Figure 4A,B). Meanwhile, we examined the expression of the key apoptosis inhibitory factors, including BCL-XL and BCL2, and pro-apoptotic factors, including BAD, BAX, CASP3 and CASP6. FTO knockdown significantly enhanced both the mRNA and protein expression of the apoptotic repressors BCL-XL and BCL2 and promoted the expression of BAX, while it had no significant effect on the expression of the other pro-apoptotic factors BAD, CASP3 and CASP6 (Figure 4C,D). ALKBH5 knockdown simultaneously promoted the mRNA expression of *BCL-XL*, *BCL2*, *BAX* and *CASP6*, did not affect the expression of *BAD* and *CASP3*, and had no apparent effect on the expression of BCL-XL, BCL2, BAD, BAX and CASP3 proteins (Figure 4C,D). In addition, METTL3 knockdown induced increased mRNA expression of *BCL-XL*, *BAX*, *CASP3* and *CASP6* but caused no noticeable changes at the protein level (Figure 4C,D). METTL14 knockdown stimulated the expression of pro-apoptotic genes BAD, BAX and CASP6 but did not affect their protein levels. WTAP knockdown inhibited BCL-XL and BCL2 expression and promoted BAD, BAX, CASP3 and CASP6 expression at both the mRNA and protein levels (Figure 4C,D). Together, our findings do not yet clarify the role of FTO in regulating apoptosis in myoblasts. On the other hand, it was demonstrated that ALKBH5 and WTAP knockdown promoted myoblast apoptosis. METTL3 knockdown inhibited myoblast apoptosis, while METTL14 knockdown likely had a neutral influence on myoblast apoptosis.

### 3.5. m^6^A Methylases Regulate Myoblast Differentiation

The formation of skeletal muscle fibers depends mainly on the fusion of myoblasts to form myotubes. Therefore, we finally investigated the role of these five m^6^A methylases on myoblast differentiation. Using the immunofluorescence of MYHC proteins to indicate myotube formation at day 3 of myoblast differentiation, we separately examined the phenotypic changes in myoblast differentiation after knockdown of each of these genes (Figure 5A). FTO, METTL3 and METTL14 knockdown all markedly inhibited myogenic differentiation in myoblasts and decreased the fusion index of multinucleated myotubes (Figure 5A,B). In contrast, ALKBH5 knockdown markedly promoted myotube fusion and generation, while WTAP knockdown had no apparent effect (Figure 5A,B). Similarly, we also examined the expression of the skeletal-myogenesis-specific factors MYOD1, MYOG, MYH3 and CKM (Figure 5C,D). Knockdown of FTO, METTL3 or METTL14 inhibited the mRNA expression of *MYOD1* and *MYOG*, while METTL3 and METTL14 knockdown did not alter *MYH3* expression (Figure 5C). ALKBH5 knockdown facilitated the mRNA expression of *MYOD1*, *MYOG*, *MYH3* and *CKM* (Figure 5C). WTAP knockdown promoted *MYOG* expression and inhibited *CKM* expression while not altering *MYOD1* and *MYH3* expression (Figure 5C). Additionally, the results of Western blotting showed that both FTO and METTL3 knockdown suppressed MYH3 protein expression, and METTL14 and WTAP knockdown suppressed MYH3 and MYOG protein levels without affecting ACTN protein levels (Figure 5D). Together, our results illustrate that FTO, METTL3 and METTL14 knockdown all inhibit myoblast differentiation, ALKBH5 knockdown promotes myoblast differentiation, whereas WTAP knockdown does not have a pronounced effect on myoblast differentiation.

To better illustrate the effect of m^6^A methylase genes on myoblast differentiation, we then constructed expression vectors for the above five m^6^A methylases. Ectopic expression of FTO contributed to *MYOD1* mRNA expression at day 3 of myogenic differentiation in myoblasts (Figure 6A), and ectopic expression of METTL3 and METTL14 also markedly contributed to mRNA expression of *MYOD1*, *MYOG*, *MYH3*, *CKM* and *MYMK* (Figure 6B,C). By contrast, ectopic expression of ALKBH5 significantly suppressed the expression of *MYH3* and *CKM* (Figure 6D). The ectopic expression of WTAP enhanced the levels of *MYH3* and *CKM* without altering the mRNA levels of *MYOD1*, *MYOG* and *MYMK* (Figure 6E). Overall, our findings indicate that FTO, METTL3 and METTL14 promote myoblast differentiation, while ALKBH5 inhibits myoblast differentiation and WTAP has a modest contribution to myogenic differentiation.

### 3.6. Phylogenetic Tree and Conserved Domains of m^6^A Methylases

Our results indicate that these five m^6^A methylases are involved in the regulation of myogenesis in bovine skeletal muscle and that each acts differently. To explore whether these roles are species-conserved, we analyzed the amino acid sequences of these five m^6^A methylases. The results clearly show that the functional domains of these five genes are well conserved in the eight analyzed species (Figure 7A–E), suggesting that they may affect skeletal muscle development in these species. This finding implies the generalizability of our results, suggesting an important role for m^6^A methylation modifications on skeletal muscle growth and development.

## 4. Discussion

In recent years, the important regulatory role of m^6^A modifications in muscle development was gradually uncovered. In addition to the abundance of reports on humans and mice, multiple studies on farm animals revealed the involvement of m^6^A modifications in regulating muscle growth and development in livestock and poultry [14,15,16,17,18,19]. However, there have still been no systematic investigations to elucidate the regulatory function of m^6^A methylases on skeletal muscle development, and few reported studies, especially on cattle. In our previous studies, we found that m^6^A modification may affect myoblast differentiation by mediating the expression of some potential targets [19], but nothing is known about the direct function of m^6^A modification on myogenesis. In the present study, the five most classical m^6^A methylases, i.e., *FTO*, *ALKBH5*, *METTL3*, *METTL14* and *WTAP*, were selected to explore their expression patterns in bovine skeletal-muscle tissues and myoblasts. Further, we used bovine skeletal-muscle myoblasts as the object of the study to discover the regulatory role of these genes in the process of skeletal myogenesis in vitro. Our findings indicate that these five m^6^A methylases have pronounced and diverse functions in regulating myoblast proliferation, apoptosis and myogenic differentiation, which can contribute to further understanding the roles of m^6^A in skeletal muscle development.

FTO has been shown to be commonly expressed in a variety of rat tissues, including muscle tissue [27]. FTO protein was highly expressed in fetal tissue and adult hypothalamus, adipose and muscle tissues [28], implying that FTO is involved in the regulation of muscle growth and development. Our results showed that *FTO* and *ALKBH5* expression were elevated during bovine skeletal-muscle growth, while *METTL3*, *METTL14* and *WTAP* expression was decreased, which clearly suggests that m^6^A methylation modifications may have a more consistent effect in skeletal muscle growth and development. This indicates that m^6^A modification is likely to have a potential role in the negative regulation of skeletal muscle development. Meanwhile, the expression levels of *METTL3* were higher and mostly variable, implying that its function may be more pronounced in the muscle development of newborn cattle. To test our conjecture, we decided to study the effect of these five m^6^A methylases on myogenesis in myoblasts in vitro. We found that all five genes follow the trend of increasing in myogenesis, especially during myogenic differentiation, indicating their potential regulatory roles in skeletal myogenesis. Consistently, increased expression of FTO during differentiation of C2C12 cells and mouse primary myoblasts [29] but decreased expression of METTL3 were observed [30]. Our results reveal that FTO knockdown promotes myoblast proliferation. Similarly, FTO was found to inhibit the proliferation of leukemic cells [31]. By contrast, the deletion of FTO was found to decrease the proliferation and differentiation of human neural stem cells (NSCs) during neurogenesis [32]. FTO knockdown also suppressed CCNA2 and CDK2 expression in 3T3-L1 preadipocytes [33]. In our results, we instead found that FTO knockdown promoted the expression of CCNA2 in myoblasts. The differences in these results may be due to differences in cell types. FTO was initially found to cause obesity [34]. Alternatively, some studies observed that the knockdown of FTO not only inhibited fat deposition in mice but also reduced muscle weight [35,36]. Subsequent in vivo and in vitro experiments verified that FTO-mediated m^6^A demethylation promotes myoblast differentiation in mice and goats [17,29]. Our results also suggest that FTO may promote myogenic differentiation by promoting the expression of MYOD1 and MYOG. Another m^6^A demethylase, ALKBH5, was identified as promoting the proliferation of tumor cells and mouse cardiomyocytes [37,38]. Similarly, the knockdown of ALKBH5 in proliferating myoblasts inhibits the expression of cell-cycle factors, and thus cell proliferation. Our results also suggest that ALKBH5 knockdown promotes myoblast apoptosis. Consistent with this, mRNA m^6^A levels were increased in mice deficient in ALKBH5, causing apoptosis with the consequent impairment of fertility [9].

Numerous studies have revealed that METTL3 promotes the proliferation of cardiomyocytes, smooth muscle cells, primary myoblasts in mice and also a variety of cancer cells [30,39,40,41,42]. However, our results showed that METTL3 knockdown promoted bovine myoblast proliferation. Therefore, further experiments may be needed to investigate the reasons for these different results. Furthermore, our results demonstrated that METTL3 knockdown inhibited myoblast apoptosis, which may be related to the fact that METTL3 knockdown mediates the m^6^A methylation of clock genes, leading to decreased apoptotic capacity [43]. Moreover, we found that METTL3 can promote the expression of the skeletal-myogenesis-specific transcription factors MYOD1 and MYOG, and thereby facilitates myogenic differentiation. Similarly, METTL3 is required for skeletal muscle regeneration in mice [44] and promotes MYOD1 expression in C2C12 cells to enhance myogenic differentiation [45]. However, some investigators found that METTL3 inhibits C2C12 differentiation [30,44]. These studies suggest that the role of METTL3 in myogenic differentiation may be complex, and subsequent studies will be carried out to conclusively determine the specific regulatory mechanisms. Our findings demonstrate that METTL14 knockdown promotes myoblast proliferation and inhibits myogenic differentiation. This finding is also identical to that of a recent study in which interference with METTL14 promoted C2C12 cell proliferation and inhibited C2C12 myoblast differentiation [15]. It was reported that WTAP can regulate the proliferation and apoptosis of a variety of cells, including 3T3-L1 preadipocytes, by controlling the expression of genes such as BCL-2, CCNA2 and CDK2 [46,47,48,49]. Our results also suggest that WTAP knockdown influences cell-cycle-factor expression to inhibit myoblast proliferation and promote apoptosis. In agreement with the results of this study’s finding that *WTAP* expression decreases in the middle and late stages of myoblast proliferation, *WTAP* expression was also observed to gradually decrease during smooth muscle cell proliferation [50].

Confusingly, both previous studies and our study showed that both m^6^A demethylase FTO and m^6^A methylase METTL3 inhibit myoblast proliferation and promote myogenic differentiation [18,29,45]. Meanwhile, FTO and ALKBH5 exhibited completely opposite effects on myogenesis in bovine skeletal myoblasts. Moreover, similar contradictory situations exist in other biological processes, such as in neurogenesis [32,39] and in hepatocellular carcinoma development [51,52]. These results suggest that the function of m^6^A methylation modifications, including in the regulation of skeletal myogenesis, is part of a complex regulatory network with the possible involvement of multiple layers.

Strikingly, we found that most of the studies on the regulation of skeletal muscle growth and development by m^6^A methylation revolve around FTO and METTL3, but there are relatively few studies on the other three methylases. Apart from our investigation here, no other relevant studies on the involvement of ALKBH5 and WTAP in skeletal myogenesis were found. In addition, the m^6^A modification is, in general, functionally interpreted by m^6^A reader proteins. The m^6^A reader proteins IGF2BP1 and IGF2BP3 were identified to inhibit C2C12 myoblast proliferation and promote myogenic differentiation [15]. However, the role of other m^6^A reader proteins and m^6^A methyltransferases in skeletal myogenesis is still waiting to be discovered to complement the regulatory network of m^6^A methylation modifications in skeletal myogenesis. We will subsequently investigate the detailed molecular mechanisms of these five m^6^A methylation enzymes in the regulation of skeletal myogenesis. We recently performed m^6^A-seq analysis during myoblast differentiation and screened several key target genes [19], which will also be experimentally validated in the future.

## 5. Conclusions

In conclusion, we systematically revealed, for the first time, the roles of five well-known m^6^A methylases in the myogenesis of bovine skeletal myoblasts. Among them, the effects of FTO and METTL3 on bovine myoblast myogenesis were found to be identical, as the knockdown of both promoted myoblast proliferation and inhibited differentiation. METTL14 knockdown promoted myoblast proliferation and inhibited differentiation but did not affect apoptosis. Moreover, FTO, METTL3 and METTL14 overexpression all promoted myoblast differentiation and ALKBH5 knockdown promoted myoblast differentiation. Meanwhile, both ALKBH5 and WTAP knockdown inhibited myoblast proliferation and promoted apoptosis. These comprehensive analyses open new perspectives for the genetic improvement and molecular breeding of beef cattle and provide a theoretical basis for studying the functional and molecular mechanisms of m^6^A methylation in regulating skeletal muscle development.

## Figures and Tables

**Figure 1 animals-12-00773-f001:**
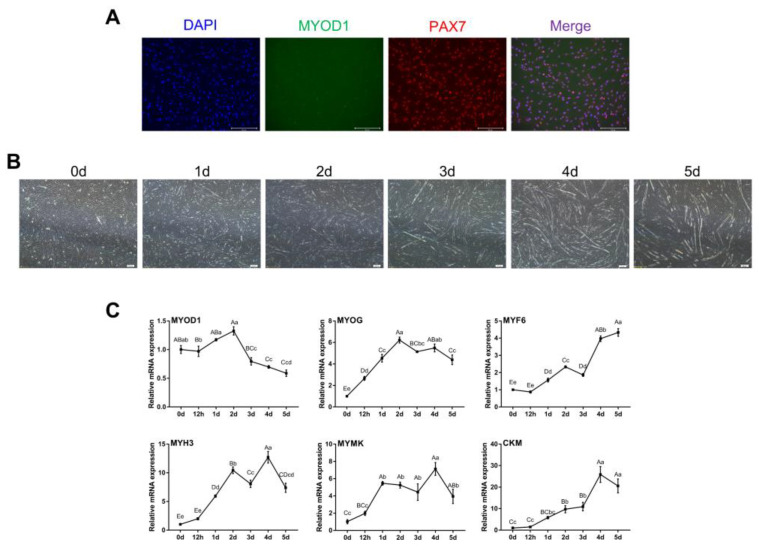
Identification of bovine skeletal myoblasts. (**A**) Identification of myoblasts by immunofluorescence based on MYOD1 (green) and PAX7 (red) expression at 48 h of growth (scale bar: 200 μm). Nuclei were counterstained with DAPI (blue). (**B**) Myotube formation on days 0, 1, 2, 3, 4 and 5 after induction of myogenic differentiation (bright field, scale bar: 200 μm). (**C**) Relative mRNA expression of skeletal-specific myogenic genes (*MYOD1*, *MYOG*, *MYF6*, *MYH3*, *MYMK* and *CKM*) during myoblast differentiation. The results were normalized to *GAPDH* levels and are presented as the means ± SDs from three independent experiments. Different capital letters indicate very significant differences (*p* < 0.01), different lowercase letters indicate significant differences (*p* < 0.05), and the same letters indicate no significant differences (*p* > 0.05) using one-way ANOVA with Tukey’s correction.

**Figure 2 animals-12-00773-f002:**
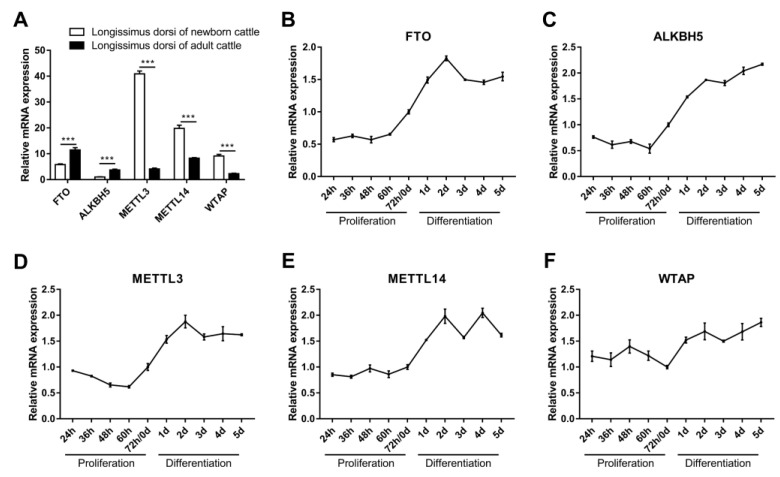
The mRNA expression of *FTO*, *ALKBH5*, *METTL3*, *METTL14* and *WTAP*. (**A**) Relative mRNA expression of *FTO*, *ALKBH5*, *METTL3*, *METTL14* and *WTAP* in longissimus dorsi muscles of newborn (1-day-old) and adult (24-month-old) cattle. (**B**–**F**) Relative mRNA expression of (**B**) *FTO*, (**C**) *ALKBH5*, (**D**) *METTL3*, (**E**) *METTL14* and (**F**) *WTAP* during proliferation and differentiation of bovine myoblasts. *GAPDH* was the normalization control. Results are presented as the means ± SDs from three independent experiments. *** *p* < 0.001, using Student’s *t* test.

**Figure 3 animals-12-00773-f003:**
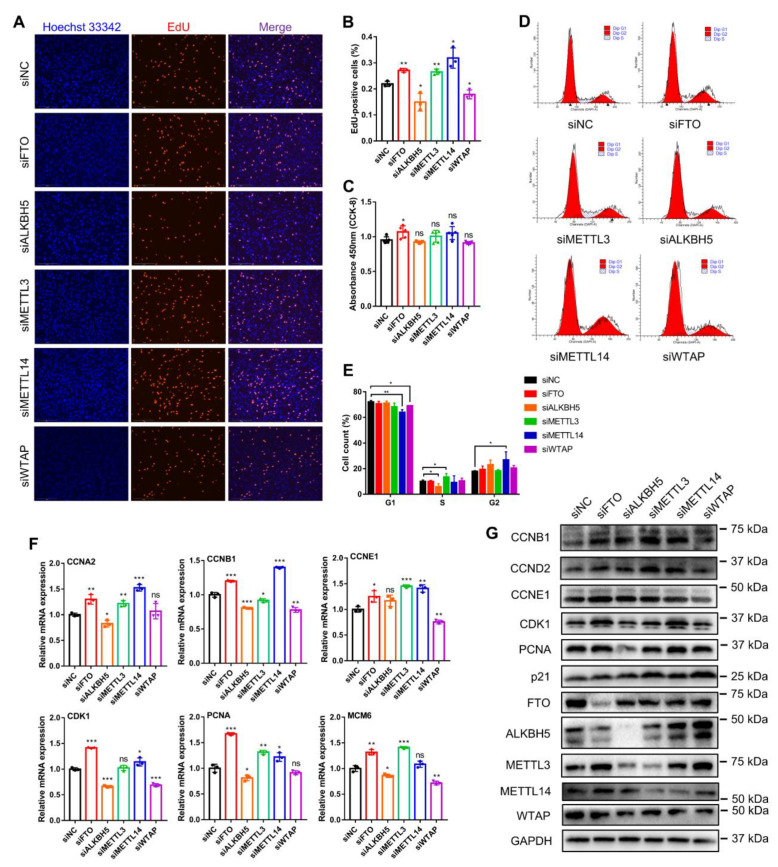
Knockdown of m^6^A methylase genes regulates bovine myoblast proliferation. The bovine myoblasts were transfected with siNC or siRNAs (siFTO, siALKBH5, siMETTL3, siMETTL14 or siWTAP) for 48 h. (**A**) The proliferation of myoblasts was detected by EdU (red) staining (scale bar: 200 μm). Nuclei were stained with Hoechst 33342 (blue). (**B**) The percentage of EdU-positive cells in (**A**) was calculated. For each group, three random microscopic fields were selected randomly. (**C**) CCK-8 assays detected the absorbance value at 450 nm to indicate the cell proliferation index. (**D**,**E**) Flow cytometry detected and analyzed the different phases of the cell cycle. (**F**) RT-qPCR was performed to detect the relative mRNA expression of cell cycle factors, including *CCNA2*, *CCNB1*, *CCNE1*, *CDK1*, *PCNA* and *MCM6*. *GAPDH* was the normalization control. (**G**) Western blot assays demonstrating the expression levels of CCNB1, CCND1, CCNE1, CDK1, PCNA, p21, FTO, ALKBH5, METTL3, METTL14, WTAP and GAPDH proteins in different treated cells. (**B**,**C**,**E**,**F**) Results are presented as the means ± SDs from three independent experiments. Using Student’s *t* test, * *p* < 0.05, ** *p* < 0.01, *** *p* < 0.001 and ns = no significance.

**Figure 4 animals-12-00773-f004:**
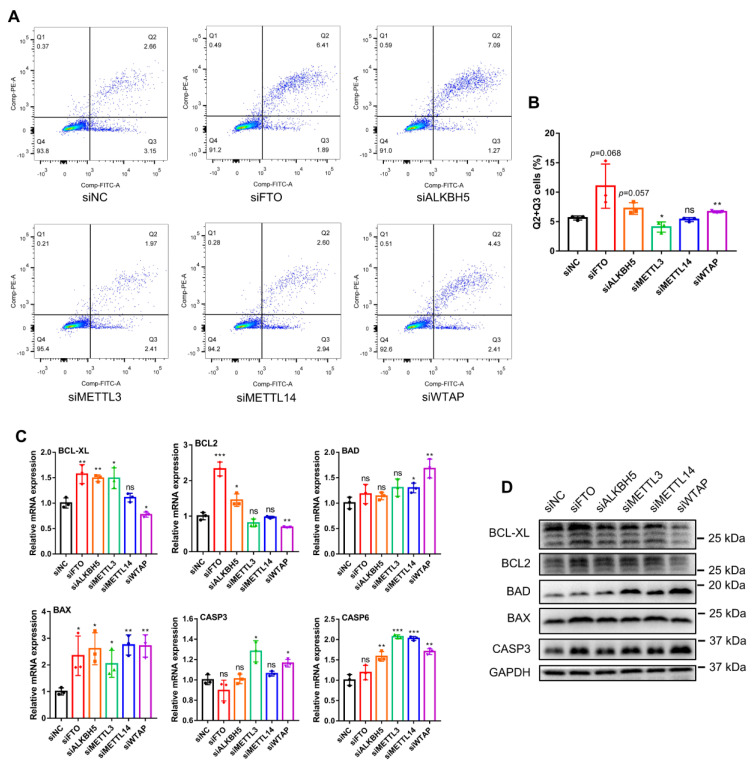
Knockdown of m^6^A methylase genes regulates bovine myoblast apoptosis. Bovine myoblasts were transfected with siNC or siRNAs (siFTO, siALKBH5, siMETTL3, siMETTL14 or siWTAP) for 36 h. (**A**,**B**) Flow cytometry detection and analysis and the percentage of apoptotic cell counts. Q1 is cell debris, Q2 is late apoptotic cells, Q3 is early apoptotic cells, and Q4 is live cells. (**C**) RT-qPCR was performed to detect the relative mRNA expression of apoptosis-related genes, including *BCL-XL*, *BCL2*, *BAD*, *BAX*, *CASP3* and *CASP6*. *GAPDH* was the normalization control. (**D**) Western blot assays demonstrating the expression levels of BCL-XL, BCL2, BAD, BAX, CASP3 and GAPDH proteins in different treated cells. (**B**,**C**) Results are presented as the means ± SDs from three independent experiments. Using Student’s *t* test, * *p* < 0.05, ** *p* < 0.01, *** *p* < 0.001 and ns = no significance.

**Figure 5 animals-12-00773-f005:**
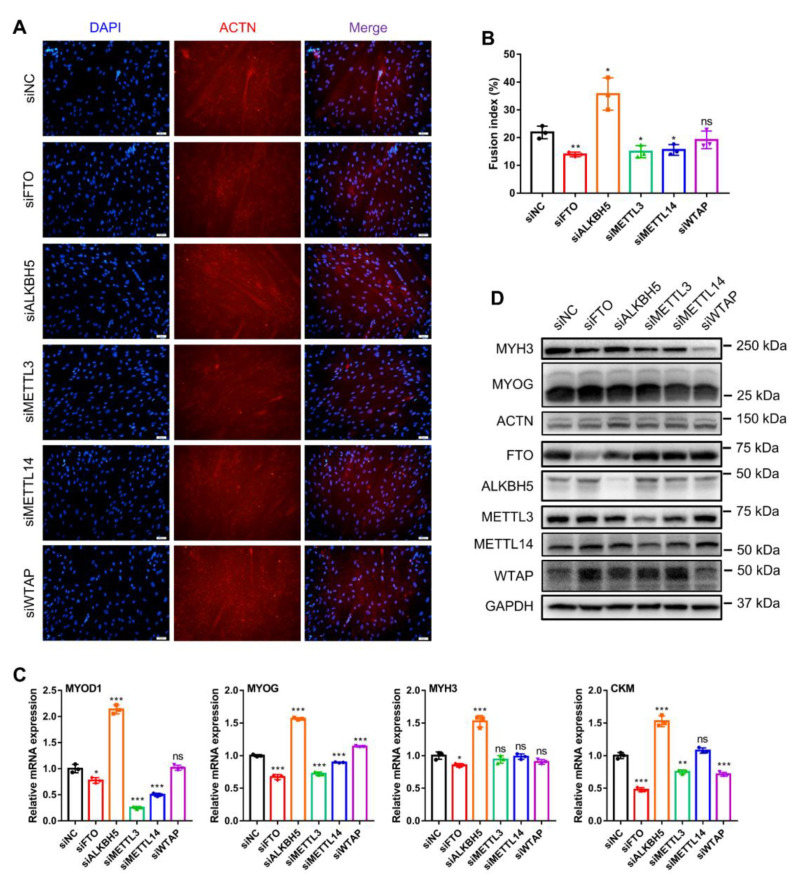
Knockdown of m^6^A methylase genes regulates bovine myoblast differentiation. Bovine myoblasts were induced for myogenic differentiation 24 h after transfection with siNC or siRNAs (siFTO, siALKBH5, siMETTL3, siMETTL14 or siWTAP). (**A**) Fluorescence staining of MYHC protein (red) indicates myotube formation on day 3 of myoblast differentiation. Nuclei were stained with DAPI (blue). (**B**) The fusion index was calculated as the percentage of nuclei in fused myotubes out of the total number of nuclei. For each group, three random microscopic fields were selected randomly. (**C**) RT-qPCR was performed to detect the relative mRNA expression of *MYOD1*, *MYOG*, *MYH3* and *CKM* on day 3 of myoblast differentiation. *GAPDH* was the normalization control. (**D**) Western blot assays demonstrating the expression levels of MYH3, MYOG, ACTN, FTO, ALKBH5, METTL3, METTL14, WTAP and GAPDH proteins in different treated cells. (**B**,**C**) Results are presented as the means ± SDs from three independent experiments. Using Student’s *t* test, * *p* < 0.05, ** *p* < 0.01, *** *p* < 0.001 and ns = no significance.

**Figure 6 animals-12-00773-f006:**
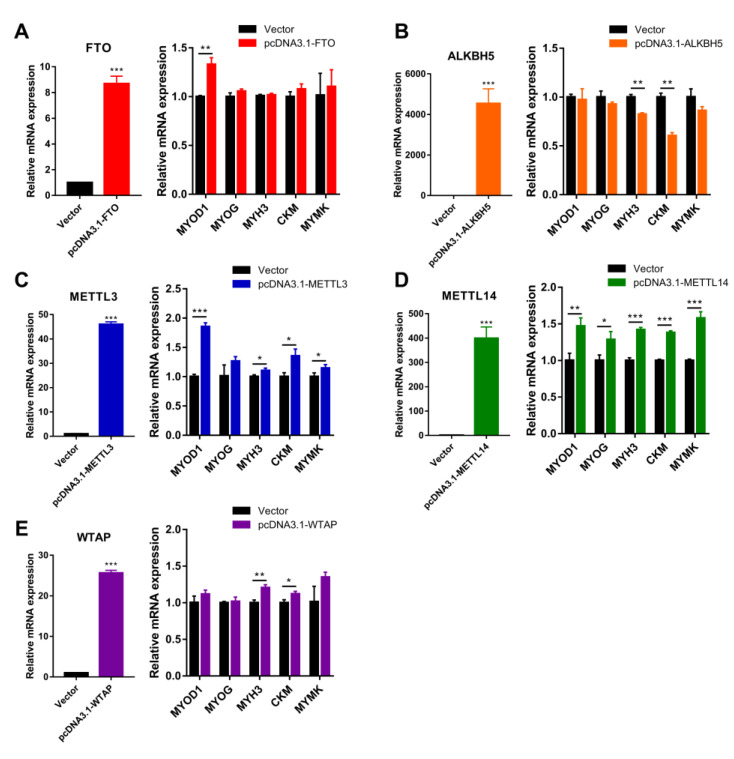
Overexpression of m^6^A methylase genes regulates bovine myoblast differentiation. Bovine myoblasts were induced for myogenic differentiation 24 h after transfection with empty vector or expression plasmids. (**A**–**E**) RT-qPCR was performed to detect the mRNA expression of (**A**) *FTO*, (**B**) *ALKBH5*, (**C**) *METTL3*, (**D**) *METTL14* and (**E**) *WTAP*, reflecting the overexpression efficiency of the expressed plasmids after transfection. Then, the relative mRNA expression of *MYOD1*, *MYOG*, *MYH3*, *CKM* and *MYMK* on day 3 of myoblast differentiation was detected. *GAPDH* was the normalization control. Results are presented as the means ± SDs from three independent experiments. Using Student’s *t* test, * *p* < 0.05, ** *p* < 0.01, *** *p* < 0.001.

**Figure 7 animals-12-00773-f007:**
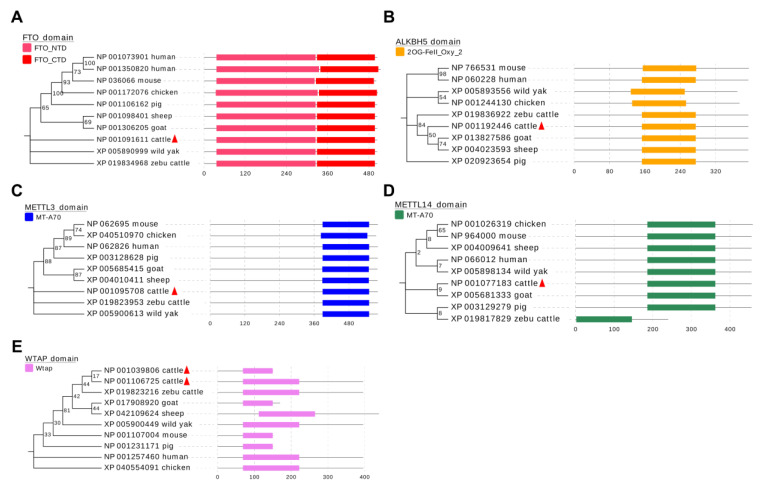
Phylogenetic tree and conserved function domains in the m^6^A methylases. (**A**) FTO, (**B**) ALKBH5, (**C**) METTL3, (**D**) METTL14 and (**E**) WTAP, visualized using MEGA7 software and the EvolView online tool.

## Data Availability

Not applicable.

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
