# Peer review of "m6A Methylases Regulate Myoblast Proliferation, Apoptosis and Differentiation"

_animals, 2022, doi:10.3390/ani12060773_

Round 1
Reviewer 1 Report
Undoubtedly an interesting and good article. Article produced in a rigorous way, the only flaw concerns the writing of the results, this last part must be rewritten by removing parts that should now be in the material and methods and discussion sections.
30-34 Even if I understand the general sense, please rewrite the sentence in a simpler way.
75 since it involves many steps it would be nice a plot or figure to resume that
173 I would put this sentence on M&M
206-208 This part belongs to the discussion
207-208 This part belongs to M&M
215 This part belongs to the discussion
218 This part belongs to the discussion
299 I would avoid terms like “curiously” in the results section
379 “We downloaded the protein sequences of FTO, ALKBH5, METTL3, METTL14 379 and WTAP from human, mouse, pig, chicken, goat, sheep, cattle and zebu cattle on the 380 NCBI website, respectively. A phylogenetic tree was constructed by MEGA7 software, 381 and the conservation of functional domains in these five proteins were subsequently ana-382 lyzed at the EvolView online website (https://evolgenius.info/evolview-v2) (Figure 7A-E)”. I would extend this part to M&M section.
Author Response
Response to Reviewer 1 Comments
Point 1: Undoubtedly an interesting and good article. Personally, I don't find it too relevant to the special issue but I would have published it as a normal article. Article produced in a rigorous way, the only flaw concerns the writing of the results, this last part must be rewritten by removing parts that should now be in the material and methods and discussion sections.
Response 1: Thanks for your comments. We have revised it.
Point 2: 30-34 Even if I understand the general sense, please rewrite the sentence in a simpler way.
Response 2: We have revised it.
Point 3: 75 since it involves many steps it would be nice a plot or figure to resume that.
Response 3: Thanks for your comment. But we don't know how to draw a general diagram of Materials and Methods.
Point 4: 173 I would put this sentence on M&M.
Response 4: We have moved this part to 2.8 Immunofluorescence in M&M.
Point 5: 206-208 This part belongs to the discussion.
Response 5: We have moved this part to Discussion.
Point 6: 207-208 This part belongs to M&M.
Response 6: We have moved this part to M&M.
Point 7: 215 This part belongs to the discussion.
Response 7: We have moved this part to Discussion.
Point 8: 218 This part belongs to the discussion
Response 8: We have moved this part to Discussion.
Point 9: 299 I would avoid terms like “curiously” in the results section.
Response 9: We have changed the word “Curiously” to “In addition”.
Point 10: “We downloaded the protein sequences of FTO, ALKBH5, METTL3, METTL14 379 and WTAP from human, mouse, pig, chicken, goat, sheep, cattle and zebu cattle on the 380 NCBI website, respectively. A phylogenetic tree was constructed by MEGA7 software, 381 and the conservation of functional domains in these five proteins were subsequently ana-382 lyzed at the EvolView online website (https://evolgenius.info/evolview-v2) (Figure 7A-E)”. I would extend this part to M&M section.
Response 10: We have moved this part to 2.9 Bioinformatics analysis in M&M.

Reviewer 2 Report
It is a manuscript with solid methodological, with minor comments. English writing needs to be improved.
It is well known that during cell differentiation of the embryo, epigenetic processes occur, such as the methylation of some genes related to muscle growth, and this process stops as the adult stage is reached. In this study, through various techniques, they were able to demonstrate that some methyltransferases such as METTL3, METTL14 and WTAP 49, promote myoblast differentiation, through the expression of the FTO gene in newborn bovines. The authors attribute that it is due to changes in the RNA sequence of these genes (methylation of adenosine on RNA). However, they did not show that these changes exist in the DNA of these genes. Is there any way to demonstrate the existence of DNA methylation in these genes?
comment:
Line 45: hot research topic
comments: look for another word, it is not common to use it
Line 112: Next,after.
comments: use another word such as, Briefly.
Line 87: comments: for gene expression analysis, it is not clear how many replicates were used.
Line 164: comments: three reps, that's pretty low, why didn't they use more reps?
Line 174: comments: in the methodology, it is not clear, because they analyzed the expression of MYOD1 and 174 PAX7 proteins. They also do not describe the full name of the protein.
Line 177: comments: in figure 1A, Merge is the control? (merge, inappropriate word).
Line 189: comments: in the methodology, the application of these genes (MYOG, MYH3, MYF6, MYMK and CKM) was not indicated, the text from line 181 to 187, should be moved to methodology.
Line 214: comments: Does high expression levels of FTO and ALKBH5 genes in adult cattle indicate slow growth, regulating demethylation?
Line 219: comments: may be more pronounced in muscle development: may be more pronounced in the muscle development of newborn cattle.
Line 220: comments: what did they find in the expression of the other five genes and what are they?
Line 488: comments: the conclusion is confused, not clear and contradictory. They should better explain the conditions where FTO, METTL3 and METTL14 inhibited myoblast proliferation and promoted myoblast differentiation.
Author Response
Response to Reviewer 2 Comments
Point 1: It is a manuscript with solid methodological, with minor comments. English writing needs to be improved.
It is well known that during cell differentiation of the embryo, epigenetic processes occur, such as the methylation of some genes related to muscle growth, and this process stops as the adult stage is reached. In this study, through various techniques, they were able to demonstrate that some methyltransferases such as METTL3, METTL14 and WTAP 49, promote myoblast differentiation, through the expression of the FTO gene in newborn bovines. The authors attribute that it is due to changes in the RNA sequence of these genes (methylation of adenosine on RNA). However, they did not show that these changes exist in the DNA of these genes. Is there any way to demonstrate the existence of DNA methylation in these genes?
Response 1: Thanks for your questions. The present study revealed the effects of these five m6A methylases (METTL3, METTL14, WTAP, FTO and ALKBH5) on myoblast proliferation, apoptosis and differentiation, but did not verify the detailed molecular mechanisms. Based on their own role in m6A methylation or demethylation, we speculate that they may regulate skeletal myogenesis by affecting the m6A methylation modification levels of some potential target mRNAs. If we were to demonstrate the presence of DNA methylation in these genes, we could investigate their changes in DNA methylation levels during skeletal myogenesis by bisulfite sequencing PCR or DNA 5-mC quantification kits.
In addition, based on our previous m6A-seq results, we have identified TET1, a DNA demethylase, as a potential target for m6A methylases to affect skeletal muscle differentiation. We have found that TET1 can also, in turn, mediate the DNA methylation levels of these m6A methylases and thus regulate their expression. These experiments are designed to validate the interplay between DNA methylation and RNA methylation in skeletal muscle differentiation and the manuscript is already in submission.
Point 2: Line 45: hot research topic
comments: look for another word, it is not common to use it
Response 2: Thanks for your comments. We considered it inappropriate for the phrase to appear here and have therefore deleted it.
Point 3: Line 112: Next,after.
comments: use another word such as, Briefly.
Response 3: We have changed the word “Next, after” to “After”.
Point 4: Line 87: comments: for gene expression analysis, it is not clear how many replicates were used.
Response 4: For the gene expression analysis in muscle and myoblasts, we used three biological replicate samples. This is also described in sections 2.1 and 2.10.
Point 5: Line 164: comments: three reps, that's pretty low, why didn't they use more reps?
Response 5: The experiments in our study, especially the gene expression analysis, were all performed in three replicate experiments, each with at least three biological replicates. The results yielded consistent conclusions for all experiments. The article showed the results of one of the experiments, which contained at least three biological replicates.
Point 6: Line 174: comments: in the methodology, it is not clear, because they analyzed the expression of MYOD1 and 174 PAX7 proteins. They also do not describe the full name of the protein.
Response 6: The methodological notes have been added in “2.8 Immunofluorescence” and the full name of the proteins have been supplemented.
Point 7: Line 177: comments: in figure 1A, Merge is the control? (merge, inappropriate word).
Response 7: In Figure 1A, Merge refers to the combined image of the photographs in the different fluorescence channels, i.e. the combined image of the DAPI, MYOD1 and PAX7 images.
Point 8: Line 189: comments: in the methodology, the application of these genes (MYOG, MYH3, MYF6, MYMK and CKM) was not indicated, the text from line 181 to 187, should be moved to methodology.
Response 8: We have moved this passage to the methodology.
Point 9: Line 214: comments: Does high expression levels of FTO and ALKBH5 genes in adult cattle indicate slow growth, regulating demethylation?
Response 9: From this result alone, we can only speculate that m6A demethylation may promote bovine skeletal muscle development. Previous studies have indicated that FTO promotes skeletal myogenesis in mice and the differentiation of sheep myoblasts. Our results also show that FTO promotes bovine myoblast differentiation, but ALKBH5 inhibits differentiation. Thus, the process by which m6A methylation regulates skeletal myogenesis may be complex.
Point 10: Line 219: comments: may be more pronounced in muscle development: may be more pronounced in the muscle development of newborn cattle.
Response 10: We have modified it. And we have moved this part to Discussion.
Point 11: Line 220: comments: what did they find in the expression of the other five genes and what are they?
Response 11: Our study was focused on five genes, METTL3, METTL14, WTAP, FTO and ALKBH5. This part of the work is to investigate the temporal expression of these five genes (METTL3, METTL14, WTAP, FTO and ALKBH5) in myoblast proliferation and differentiation.
Point 12: Line 488: comments: the conclusion is confused, not clear and contradictory. They should better explain the conditions where FTO, METTL3 and METTL14 inhibited myoblast proliferation and promoted myoblast differentiation.
Response 12: The conclusions have been revised.

Round 2
Reviewer 2 Report
The indicated corrections were made, I consider that the document has the potential to be accepted under these conditions. I have these observations.
233. review
238. review
245. (Figure 2B-F) or (Figure 2B, F), uniform
248. (Figure 2B-F) or (Figure 2B, F?, uniform
296. remove the point.
317. (Figure 4C,D) or (Figure 4C-D)?, uniform
322. (Figure 4C,D) or (Figure 4C-D)? uniform
Author Response
Point 1: 245. (Figure 2B-F) or (Figure 2B, F), uniform
Response 1: It has been uniformly revised to "(Figure 2B-F)".
Point 2: 248. (Figure 2B-F) or (Figure 2B, F?, uniform
Response 2: It has been uniformly revised to "(Figure 2B-F)".
Point 3: 296. remove the point.
Response 3: We have removed it.
Point 4: 317. (Figure 4C,D) or (Figure 4C-D)?, uniform
Response 4: It has been uniformly revised to "(Figure 4C-D)".
Point 5: 322. (Figure 4C,D) or (Figure 4C-D)? uniform
Response 5: It has been uniformly revised to "(Figure 4C-D)".
